# PBMCs as Tool for Identification of Novel Immunotherapy Biomarkers in Lung Cancer

**DOI:** 10.3390/biomedicines12040809

**Published:** 2024-04-05

**Authors:** Caterina De Rosa, Francesca Iommelli, Viviana De Rosa, Giuseppe Ercolano, Federica Sodano, Concetta Tuccillo, Luisa Amato, Virginia Tirino, Annalisa Ariano, Flora Cimmino, Gaetano di Guida, Gennaro Filosa, Alessandra di Liello, Davide Ciardiello, Erika Martinelli, Teresa Troiani, Stefania Napolitano, Giulia Martini, Fortunato Ciardiello, Federica Papaccio, Floriana Morgillo, Carminia Maria Della Corte

**Affiliations:** 1Department of Precision Medicine, University of Campania Luigi Vanvitelli, 80131 Naples, Italy; caterina.derosa1@unicampania.it (C.D.R.); concetta.tuccillo@unicampania.it (C.T.); luisa.amato@unicampania.it (L.A.); annalisaariano98@outlook.it (A.A.); gaetano.diguida@studenti.unicampania.it (G.d.G.); gennaro.filosa@unicampania.it (G.F.); alessandradiliello@gmail.com (A.d.L.); erika.martinelli@unicampania.it (E.M.); teresa.troiani@unicampania.it (T.T.); stefania.napolitano@unicampania.it (S.N.); giulia.martini@unicampania.it (G.M.); fortunato.ciardiello@unicampania.it (F.C.); floriana.morgillo@unicampania.it (F.M.); 2Institute of Biostructures and Bioimaging, National Research Council, 80145 Naples, Italy; francesca.iommelli@ibb.cnr.it; 3Department of Pharmacy, School of Medicine, University of Naples Federico II, 80138 Naples, Italy; giuseppe.ercolano@unina.it (G.E.); federica.sodano@unina.it (F.S.); 4Department of Experimental Medicine, University of Campania Luigi Vanvitelli, 81100 Caserta, Italy; virginia.tirino@unicampania.it; 5U.P. Diagnostica Citometrica e Mutazionale, A.O.U. Vanvitelli, Università degli Studi della Campania, 80138 Naples, Italy; 6Hospital “Martiri Di Villa Malta”, 84087 Sarno, Italy; floracimmino81@gmail.com; 7Division of Gastrointestinal Medical Oncology and Neuroendocrine Tumors, European Institute of Oncology (IEO), IRCCS, 20141 Milan, Italy; davide.ciardiello@unicampania.it; 8Department of Medicine, Surgery and Dentistry, “Scuola Medica Salernitana”, University of Salerno, 84084 Baronissi, Italy; fpapaccio@unisa.it

**Keywords:** PBMC, cGAS-STING, biomarker

## Abstract

Background: Lung cancer (LC), including both non-small (NSCLC) and small (SCLC) subtypes, is currently treated with a combination of chemo- and immunotherapy. However, predictive biomarkers to identify high-risk patients are needed. Here, we explore the role of peripheral blood mononuclear cells (PBMCs) as a tool for novel biomarkers searching. Methods: We analyzed the expression of the cGAS-STING pathway, a key DNA sensor that activates during chemotherapy, in PBMCs from LC patients divided into best responders (BR), responders (R) and non-responders (NR). The PBMCs were whole exome sequenced (WES). Results: PBMCs from BR and R patients of LC cohorts showed the highest levels of STING (*p* < 0.0001) and CXCL10 (*p* < 0.0001). From WES, each subject had at least 1 germline/somatic alteration in a DDR gene and the presence of more DDR gene mutations correlated with clinical responses, suggesting novel biomarker implications. Thus, we tested the effect of the pharmacological DDR inhibitor (DDRi) in PBMCs and in three-dimensional spheroid co-culture of PBMCs and LC cell lines; we found that DDRi strongly increased cGAS-STING expression and tumor infiltration ability of immune cells in NR and R patients. Furthermore, we performed FACS analysis of PBMCs derived from LC patients from the BR, R and NR cohorts and we found that cytotoxic T cell subpopulations displayed the highest STING expression. Conclusions: cGAS-STING signaling activation in PBMCs may be a novel potential predictive biomarker for the response to immunotherapy and high levels are correlated with a better response to treatment along with an overall increased antitumor immune injury.

## 1. Introduction

During the past years several new therapies have been developed for lung cancer (LC). Both non-small (NSCLC) and small lung cancer (SCLC) subtypes are currently treated with a combination of chemo- and immunotherapy, with variable duration of responses. In almost all patients, at variable time points disease progression occurs [1,2]. Thus, we still need novel biomarkers and novel potential combinations [3,4,5]. Nowadays, the overactivation of the PD-L1/PD-1 axis represents a good target and biomarker for cancer treatment, but it is not sufficient to identify immunotherapy-responsive patients and it is not able to reflect the dynamicity of immune response. In this respect, there is a need to use accurate predictive biomarkers for an early identification of responder (R) and non-responder (NR) patients. In this study, we hypothesized that certain biological parameters, which can be potentially detected in peripheral blood, could be used as surrogate biomarkers of a predictive tumor response [6,7]. Our group and others have demonstrated that activation of innate immune pathways in cancer cells, such as canonical cGAS/STING signaling [5], is a good predictor of antitumor immune response [3] and that exceptionally long R LC patients may harbor germline variants in DDR genes [8]. In the present work, we used peripheral blood immune cells (PBMCs) as a tool for investigating the innate immune response and as a source for genetic testing, proposing an innovative strategy for monitoring and predicting immunotherapy response.

## 2. Materials and Methods

### 2.1. Treatment and Study Assessments

We enrolled patients with a diagnosis of NSCLC or SCLC receiving one of the following treatments: chemotherapy (cisplatin) and/or an anti-PD-L1 antibody (atezolizumab, durvalumab). The patients were grouped as follows: best responders (BR) were considered as patients with controlled disease lasting more than 12 months in NSCLC and more than 6 months in SCLC, that were clinically relevant compared with registration clinical trials [9,10,11]; responders (R) were defined as patients achieving at least stable disease (SD) at first radiological assessment, while non-responders (NR) were defined as patients showing progression of disease (PD) as the best response.

### 2.2. Peripheral Blood Mononuclear Cells (PBMC) Isolation: Step-By-Step Method 

Human samples were collected after obtaining a written informed consent from patients in accordance with the Declaration of Helsinki. The protocol for the use of these samples for research purposes was approved by the Ethics Committee of the University of Campania “Luigi Vanvitelli”, Naples (*n*. 280 on 16 May 2020). Lung cancer patients’ blood serum was collected by centrifugation and stored immediately at −20 °C or prepared for the ELISA assay kit for the detection of CXCL10 and CCL5 (Invitrogen™, Santa Clara, CA, USA). Subsequently, peripheral blood was collected in BD vacutainer spray-coated K2EDTA tubes (BD, Franklin Lakes, NJ, USA) as described in our previous studies [8,12,13]. Within 2 h of blood draw, PBMCs were isolated by using Lymphosep (Aurogene Srl, Rome, Italy) gradient centrifugation. Furthermore, to eliminate the contamination of red blood cells (RBCs) in the PBMC samples, we suspended the PBMC pellets in the RBC lysis solution (Invitrogen™, Santa Clara, CA, USA). After several washes with PBS, PBMCs were transferred into TRIsure™ (Bioline, Meridian Bioscience, Memphis, TN, USA) and stored at −80 °C until use or immediately resuspended in RPMI 10% FBS containing human autologous serum (10%), ultraglutamine I (1%), penicillin and streptomycin (1%). After isolation, immune cells were cultured in presence of beads coated with anti-CD3 and anti-CD28 (Life Technologies, Carlsbad, CA, USA) at a ratio of 1 bead per 10 cells and IL-2 (Miltenyi Biotech, Bergisch Gladbach, Germany) at a concentration of 20 U/mL. After 24 h of culture, cisplatin 0.5 µM or DNA-PK-I 2 µM concentration was added for a further 72 h. Culture supernatants were collected for further analysis. Representative images shown in Figure 1 report a detailed scheme of the PBMC isolation method step-by-step.

### 2.3. Enzyme-Linked Immunosorbent Assay (ELISA)

The protein expression levels of extracellular cytokines and chemokines (CXCL10 and CCL5/RANTES) were measured in the blood serum of lung cancer patients, using the Human RANTES ELISA Kit (cat. No. EHRNTS) for CCL5/RANTES and Human IP-10 ELISA Kit (cat. No. KAC2361) for CXCL10 (Invitrogen; Thermo Fisher Scientific, Inc., Waltham, MA, USA). The amounts of proteins were measured in clear flat-bottom 96-well plates in accordance with the manufacturers’ instructions. The optical density was determined using the Infinite M Plex (Tecan, Männedorf, Switzerland) microplate reader set to 450 and 550 nm. The readings at 550 nm were subtracted from the 450 nm reading to correct for optical imperfections in the plate. The concentrations (pg/mL) of the different proteins in each sample were determined by interpolating from the absorbance value (*Y* axis) to protein concentration (*X* axis) using the standard curve (Appendix A). Each experiment was performed in duplicate. Data are expressed as mean ± SD.

### 2.4. RNA Extraction and cDNA Synthesis 

Total RNA was obtained from cell lines using TRIsure reagent (BIO-38033, Meridian Bioscience, Memphis, TN, USA), according to the manufacturer’s protocol. Briefly, cells (5 × 10^6^) were lysed with 1 mL of TRIsure and incubated for 5 min at RT. Chloroform (0.2 mL) was added and samples were shaken vigorously by hand for 15 sec and incubated for 3 min at RT. Samples were then centrifuged at 12,000× *g* for 15 min at 4 °C. The samples separated into a pale green, organic phase, an interphase, and a colorless upper aqueous phase. The aqueous phase was removed for RNA extraction and transferred to another tube. RNA was precipitated by mixing with cold isopropyl alcohol (0.5 mL) for 2 h at −80 °C. Samples were centrifuged at 12,000× *g* for 10 min at 4 °C and washed twice with 75% ethanol. Total RNA was suspended in 50 μL of RNase-free water. The purity and concentration of RNA were determined by OD 260/280 readings using the Nanodrop 2000 spectrophotometer (Thermo Fisher Scientific, Waltham, MA, USA). After RNA extraction, cDNA was generated from 500 ng of total RNA using a SensiFAST cDNA Synthesis Kit (BIO-65053, Meridian Bioscience, Memphis, TN, USA) under the following conditions: 25 °C for 10 min, 42 °C for 15 min, 85 °C for 5 min.

### 2.5. Gene Expression Analysis by Quantitative qRT-PCR

mRNA expression levels of *STING*, *cGAS*, *CXCL10* and *CCL5* genes were evaluated by qRT-PCR with a QuantStudio 7-Flex (Applied Biosystems by Life Technologies, Monza, Italy) using the SensiFAST SYBR Hi-ROX Kit (BIO-92005, Meridian Bioscience, Memphis, TN, USA) under the following conditions: 50 °C for 2 min (stage 1) followed by a denaturation step at 95 °C for 10 min (stage 2) and then 40 cycles at 95 °C for 15 s and 60 °C for 1 min (stage 3). All samples were run in duplicate, in 20 μL reactions and relative expression of genes was determined by normalizing to *18S*, used as an internal control gene; to calculate relative gene expression in value we used the 2^−ΔCt^ or 2^−ΔΔCt^ method. Table 1 shows the list of primer sequences used for qRT-PCR.

### 2.6. Next-Generation Sequencing (NGS)

NGS analysis, alignment and filtering analysis were performed by the AMES service (Casalnuovo, Italy). Germline mutations were first filtered as a major allele frequency (MAF) of 0.01% in the database Exome Aggregation Consortum (ExAC-EAS). We then performed an analysis of germline variants which led to loss of function (LoF): (1) gain of stop codon, (2) loss of initiation codon, (3) frameshift, (4) deletion of single exon, (5) missense change and inframe del predicted by in silico tools as deleterious (poliphen-2 and SIFT) and reported in ClinVar database as pathogenic or uncertain significance variants.

### 2.7. Generation of 3D Spheroid Culture

NCI-H661 (ATCC CAT#HTB-183; RRID: CVCL_1577) and H460 (ATCC CAT#HTB-177; RRID: CVCL_0459) were maintained in RPMI 1640 (R8758, Sigma-Aldrich, Merck, Darmstadt, Germany) supplemented with 10% FBS (Sigma-Aldrich, Merck, Darmstadt, Germany) and 1× penicillin-streptomycin (P0781, Sigma-Aldrich, Merck, Darmstadt, Germany) in a humidity-controlled environment (37 °C, 5% CO_2_). NCI-A549 (ATCC CAT#CCL-185; RRID: CVCL_0023) was maintained in DMEM (D5030, Sigma-Aldrich, Merck, Darmstadt, Germany) supplemented with 10% FBS and 1× penicillin-streptomycin in a humidity-controlled environment (37 °C, 5% CO_2_). Cell lines were grown in standard culture conditions, harvested and dissociated into single cell suspensions for spheroid generation. Tumor spheroids were generated by seeding 10^4^ cells per well in ultra-low attachment (Corning, New York, NY, USA) round-bottom 6-well plates in the same culture media used for the monolayer cultures. Plates were incubated at 37 °C in 5% CO_2_ and spheroids were maintained by performing medium replenishments every 3–4 days. Cancer cells started to form spheroids at day 1 and well-formed spheroids were seen at day 7.

### 2.8. Co-Localization of 3D Tumor Spheroids and LC Patient-Derived PBMCs by Immunofluorescence (IF)

Seven days after cell seeding, the spheroids were ready for co-culture. Co-cultures were started by adding 5 × 10^5^ LC patients-isolated PBMCs per well pre-treated with DNA-PK-I (2 µM) or cisplatin (0.5 µM) for 72 h. Before co-culture, 50 µM CellTracker Deep Red dye (Cat. No. C34565, Invitrogen, Carlsbad, CA, USA) was added to PBMC culture medium for 45 min and 10 µM CellTracker Blue CMAC (7-amino-4-chloromethyl coumarin) (Cat. No. C2110, Invitrogen, Carlsbad, CA, USA) was added to the spheroid culture medium for 45 min. After the staining, the PBMCs were added to spheroids and the co-cultures were performed for 72 h. After 72 h, the fluorescence was imaged using a Cell Discoverer 7 microscope (ZEISS Olympus, Nikon, Tokyo, Japan) at the facility center of CEINGE Biotecnologie Avanzate (Napoli, Italy). The quantification of the mean fluorescence intensity (MFI) was performed with ImageJ Software version 1.52e.

### 2.9. Flow Cytometry 

After isolation, PBMCs from LC patients were stained and analyzed using standard flow cytometry approaches. Single cell suspensions from all samples were stained with Live Dead Blue (Invitrogen, Carlsbad, CA, USA, 1:400 in PBS) for 10 min at 4 °C, washed with FACS buffer (1% FBS in PBS) and then stained with cell surface marker antibodies for 45 min at 4 °C. For STING and Granzyme B staining, cells were fixed and permeabilized with FIX & PERM™ Cell Permeabilization Kit (Thermo Fisher Scientific, Waltham, MA, USA).

Antibodies used were anti-human CD3 (UCHT1, SK7), CD4 (L3T4, SK3), CD8 (SK1), CD14 (MφP9), CD56 (NCAM16.2), CD19 (SJ25C1), CD16 (3G8) from BD Bioscience; anti-human STING (#90947) from Cell Signaling Technology, Danvers, MA, USA; and anti-human Granzyme B (REA226) from Miltenyi Biotech (Bergisch Gladbach, Germany). For STING labeling, a secondary FITC-labeled rabbit monoclonal antibody (Abcam, Cambridge, UK) was added at 4 °C for 30 min in the dark.

Labeled cells were washed and resuspended in FACS buffer for flow cytometric analysis on a BD Fortessa with FACS Diva software (BD Biosciences, Franklin Lakes, NJ, USA) and analyzed using FACS Diva software, version 8.0. Negative gating was based on a fluorescence-minus-one (FMO) strategy.

### 2.10. Multiplex Cytokine Assay 

The concentrations of various cytokines in cell-free culture supernatants were determined using custom human LEGENDplex™ analyte assay kits (Biolegend Inc., San Diego, CA, USA). The assay was performed according to the manufacturer’s instructions. Samples were acquired on a Bricyte E6 (Mindray, Shenzhen, China). Data were analyzed using FlowJo software (TreeStar V.10).

### 2.11. Statistical Analysis 

Results are expressed as the mean ± SEM or SD. Two-group analyses were performed using an unpaired *t*-test. Three or more groups with one independent variable were analyzed using a one-way ANOVA test. Three or more groups with two independent variables were analyzed using a two-way ANOVA test. The Pearson correlation coefficients and the corresponding *p*-values were calculated using Prism 8 (GraphPad Software, San Diego, CA, USA). Analyses were performed using the Prism 8 (GraphPad Software, San Diego, CA, USA) software. All tests were two-tailed and a *p*-value < 0.05 was considered to indicate statistical significance. All the experiments were repeated a minimum of three times independently to ensure reproducibility.

## 3. Results

### 3.1. Study Population, Patient Characteristics and Clinical Responses

In this study, 28 patients with a diagnosis of NSCLC or SCLC who had undergone at least two cycles of PD-L1 inhibitors between 2022 and 2023 were enrolled. Clinical and demographic variables for lung cancer cases are detailed in Table 2. Baseline patient characteristics were as follows: 13 patients had adenocarcinoma (NSCLC) and 15 had SCLC. In total, 3 patients (10.7%) received PD-L1 inhibitors as single agent first-line treatment; 25 patients received PD-L1 inhibitors combined with chemotherapy (89.3%). The median age of the patients was 66.34 years. In total, 7 (25%) patients achieved a total response, 13 (46.4%) remained stable (SD), and 8 (28.6%) showed PD.

### 3.2. Association of Cytosolic DNA Sensor cGAS/STING Levels in Peripheral Blood with Response to Anti-PD-L1 Blockade

Our group previously reported that the activation of the STING pathway predicts the effectiveness of immunotherapy response in LC [5]. Moreover, CXCL10 and CCL5 are among the two most well-known secreted chemokines activated by the cGAS/STING pathway [14]. Due to limited access to biopsies, particularly from SCLC patients and metastatic NSCLC patients, we aimed to investigate the potential of STING pathway activation through the evaluation of mRNA expression of cGAS-STING and downstream chemokines CXCL10/CCL5 in total PBMCs from LC patients and their correlation to the clinical response to anti-PD-L1 blockade. In Figure 2, the mRNA expression levels of *STING*, *cGAS*, *CXCL10* and *CCL5* in such blood cells were significantly lower (*p* < 0.0005) in the NR cohort as compared to BR. The decrease in mRNA fold change was associated with a poor response to anti-PD-L1 therapy.

Furthermore, as shown in Table 3, analysis of expression levels between samples showed positive correlation between the mRNA expression of *STING* and *CXCL10* (Pearson r = 0.6705; *p* = 0.0339) in the BR cohort. In R cohort, *STING* mRNA expression positively correlated with *cGAS* (Pearson r = 0.7053; *p* = 0.0227), while *cGAS* mRNA expression was also significantly correlated with *CXCL10* (Pearson r = 0.8643; *p* = 0.0013) and *CCL5* (Pearson r = 0.8796; *p* = 0.0008). Furthermore, in BR and R cohorts, *CXCL10* and *CCL5* correlated with each other (Pearson r = 0.8777; *p* = 0.0008 and Pearson r = 0.9051; *p* = 0.0003, respectively). In the NR cohort, *STING* and *cGAS* did not show significant correlations with *CXCL10* or *CCL5* mRNA expression. No significant correlations were found between *CXCL10* and *CCL5* in the NR cohort.

These data suggest that cGAS-STING activation is present in PBMCs of patients with clinical response to anti-PD-1/PD-L1 immunotherapy, while it is not detectable in NR patients. STING activation is known to be a mediator of PD-1 expression [5], supporting the efficacy of anti-PD-1/PD-L1 agents. Since *cGAS* is not clearly correlated with *STING, CXCL10* and *CCL5* mRNA in BR patients, we hypothesize that in this subgroup STING may be activated by other immune mediators beyond the canonical cGAS-mediated signal. Thus, these data support a further prospective study in a larger population of patients.

### 3.3. Determination of STING-Related Serum Cytokines in Patients with Lung Cancer under Anti-PD1/PD-L1 Therapy

CXCL10/CCL5 serum markers are among the most important published evidence of a potential association with clinical outcome in NSCLC patients receiving PD-L1 inhibitors. The evidence was gathered from 624 original publications identified in PubMed from Schindler H. et al. [15]. Recent evidence showed that DNA damage and accumulation of cytosolic dsDNA leads to activation of the cGAS-dependent STING pathway in lung adenocarcinoma cells. Specifically, activated T lymphocytes induced DNA damage and activated the STING pathway in LC cells [16]. Although various studies on predictive markers in the use of PD-1/PD-L1 inhibitors are in progress, only PD-L1 expression levels in tumor tissues are currently used. Here, we investigated whether baseline serum levels of CXCL10/CCL5 could represent a good surrogate for the treatment response of patients with advanced NSCLC or SCLC treated with PD-L1 inhibitors. Figure 3 shows a significant reduction of both CXCL10 and CCL5 (*p* < 0.0001) in NR serum as compared to BR cohorts. However, analysis shown in Appendix A results did not find any significant correlation between serum levels of these proinflammatory chemokines in BR and R patients. These data may be explained by the fact that cytokine/chemokine baseline levels may be highly variable among patients, thus needing further validation in more extensive sample size studies to find relevant cutoff values. Probably, integration with other features would anyway be necessary for predicting immunotherapy outcome and monitoring the changes over time of these values may be more useful in future studies to better define the cutoff for clinical implications.

### 3.4. Germline Variants in DNA Damage Response (DDR) Genes Potentially Implicated in Immune Response

A recent publication showed that about 15% of LC patients harbor a germline pathogenic variant, with a significant enrichment in DDR genes [17]. Pathogenetic variants of DDR genes reduce the ability to effectively repair single- and double-strand breaks (SSB and DSB) after DNA damage, thus increasing cancer cell sensitivity to certain therapies. However, the effect of these variants on immune cells is still uncharacterized. Indeed, in a previous publication, we analyzed three long immunotherapy responder patients and found that they presented rare germline variants in DDR genes [8]. Independently from their putative role in cancer induction/promotion, as immune cells are the main effectors of immunotherapy, we aimed at investigating the possible presence of these variants in a larger patient cohort. Moreover, based on mRNA correlation results in NR patients, we looked for further targets in the NR cohort by performing germinal analysis. Therefore, all patients underwent germline genetic testing focusing on a panel of DDR genes according to our previous publication [8]. Interestingly, we identified heterozygous germline LoF variants in DDR genes (Table 4). In particular, we found germline *POLE*, *POLD1*, *RAD51 paralog B* (RAD51B), checkpoint kinase 1 (*CHEK1*) and *ATM* pathogenic variants in BR and R SCLC patients, whereas NSCLC BR/R patients harbored germline pathogenic variants in *BARD1* and *ATM* (Figure 4) [8,18,19,20,21]. Importantly, no germline variants in the DDR genes panel were found in the NR cohort. This trend, if later validated in a larger cohort of patients, may suggest that deleterious DDR variants in the PBMCs of LC patients might have an impact on their clinical response to immunotherapy. In contrast, LC patients with no deleterious DDR variants may be less responsive to treatments.

### 3.5. Effect of DNA-Dependent Protein Kinase (DNA-PK) Inhibitor on Cytosolic DNA Sensor cGAS/STING in Peripheral Blood of LC with Different Response to Anti-PD-L1 Blockade

As we did not find any functional alterations of DDR proteins in patients less responsive to immunotherapy, we hypothesized that their more functional state compared to BR might mediate immunotherapy resistance. Therefore, we tested a DDR inhibitor, DNA-PK-I, on PBMCs from LC patients according to the clinical responses to anti-PD1/PD-L1 therapy. As shown in Figure 5, NR patients respond to DNA-PK-I with a significant increase in the cGAS/STING pathway, suggesting that inhibition of this pathway is an important requirement for the activation of mediated immunity in PBMCs. The strong increase in *cGAS/STING* mRNA levels in response to DNA-PK inhibition in NR patients may be explained by the effect on perturbation by the drug on DDR machinery. On the other hand, because the DDR machinery is genetically altered in the BR and R cohorts, this may represent the intrinsic mechanism that also causes DNA damage-dependent cGAS/STING pathway activation in immune cells.

Taken together, these results suggest that PBMCs from LC patients may be a tool to study the activation of immune pathways with novel drugs such as DDR inhibitors that are under clinical investigation for LC; also, these results may be of interest if correlated with the presence of germline DDR gene alterations as potential novel biomarkers.

### 3.6. Antitumoral Activity and Innate Immune Cells Effect of DNA-PK Inhibitor

Here, we aimed to evaluate the functional effect of cisplatin, the standard of care for LC patients in combination with immunotherapy, on LC patient-derived PBMCs. Specifically, we tested the applicability of PBMCs as an in vitro tool to study and reproduce the effect of cisplatin on immune cells in the tumor microenvironment. Additionally, we evaluated the effect of DNA-PK-I as a potential candidate drug to activate PBMCs based on a similar effect on DNA damage and DDR machinery perturbation in a chemotherapy-like way. We propose a 3D co-culture model of different types of lung cancer cell lines (H460, H661 and A549) and LC patient-derived PBMCs. As showed in Figure 6, untreated PBMCs from LC patients were not able to access the compact 3D tumor structure. Interestingly, cisplatin-treated immune cells infiltrated the spheroids at day 5 and the spheroid’s 3D structure was destroyed (Figure 6). We also used DNA-PK-I pre-treated PBMCs as controls for DNA damage induction since it has been widely demonstrated that DNA-damaging therapies induce a DNA sensor STING-dependent antitumor immune response [5,22]. Quantitative analysis showed that both cisplatin and DNA-PK-I were able to significantly increase the red/blue intensity ratio in H460 (Figure 6A), H661 (Figure 6B) and A549 (Figure 6C), indicating an increased infiltration ability towards the 3D tumor spheroids. Taken together, these results demonstrated that 3D co-culture models of tumor spheroids and PBMCs represent a valid surrogate of mouse models for functional in vitro prediction of therapy-induced immune system activation and strengthen the rationale for more tailored mechanistic studies in animal models. Moreover, the results obtained with DNA-PK-I potentially open up new treatment options for NR patients to be tested in a clinical setting.

### 3.7. Subpopulation Analysis of STING Expression in LC Patient-Derived PBMCs 

Since PBMCs are extremely heterogeneous, to further analyze which subset was more implicated in cGAS/STING activation, we examined PBMCs derived from LC patients from the BR, R and NR cohorts by flow cytometry and determined which gated subpopulation displayed a higher STING expression. CD8+ T cell subsets were significantly enriched in the BR and R cohorts, as shown in Figure 7A. In addition, the BR cohorts also showed a higher percentage of NK cells compared to the R and NR cohorts. Among the PBMC gated subsets derived from LC patients, the percentage of STING-positive cells was significantly higher in CD8+ T cells. These results indicate that cytotoxic T lymphocytes (CTLs) and natural killer (NK) cells are the predominant subset among lymphocytes in BR PBMCs derived from LC patients. In addition, CTLs are the major contributors to STING expression in PBMCs derived from LC patients. To further confirm our findings, we assessed the cytotoxic activity of innate immune cells by determination of cytokine and cytotoxic granule protein levels. In particular, we focused on the secretion of granzyme A, granzyme B, perforin and granulysin, which are mainly secreted by CTLs and NK cells. In addition, they are highly associated with the innate immune response to various intracellular pathogens as well as tumors [8,23,24]. Therefore, we quantified these mediators in culture supernatants of LC patient-derived PBMCs by multiplex analysis. As shown in Figure 7B, we found that the BR cohort highly secreted granzyme A, granzyme B, perforin and granulysin as compared to the NR cohort. Taken together, these results suggest that cGAS/STING activation in CTLs, which are most known for releasing proteins contained within the cytolytic granules, positively influences the antitumor cytotoxic activity of innate immune cells.

## 4. Discussion

Despite a large number of candidate biomarkers under clinical investigation to select patients for immunotherapy, only a limited number of molecular assays based on PD-L1 expression are currently approved by the FDA. In particular, these noninvasive tests have been clinically validated to identify patients more likely to benefit from single-agent anti-PD-1/PD-L1 therapy. However, due to the complex interplay between immune reaction and tumor biology, a more accurate analysis will be relevant to predict patient response to immune-targeted therapy. Indeed, accurate prediction of clinical benefit may require combination of multiple tumor and immunological response markers, such as protein expression, genomics and transcriptomics. For example, in syngeneic mouse models of cancer, an antitumor immunity effect has been observed following intratumoral administration of a STING agonist [25]. Furthermore, the effects of radiotherapy [26], checkpoint inhibitors [5] and, recently, PARP inhibitors [27], are highly dependent on the presence of STING, and anti-cancer effects are enhanced when combined with STING agonists. In a cancer setting, it is obviously necessary to study immune status and activation within the cancer site, but the peripheral immune system can also provide important information, particularly on intrinsic function and immune response capacity. Considering the low availability of tissue material from LC patients, in this work we provided evidence of using PBMCs as a new tool for biomarker study. Further investigations will be carried out to evaluate the applicability of these results to transform SCLC that is derived from NSCLC. Even if we recognize the potential limit of inter-patient variability, with this pivotal study we had a coherent result in terms of immune system activation and clinical response to anti-PD-L1 in PBMCs by testing STING-related markers. Moreover, PBMCs allowed us also to perform functional studies that may be useful to predict immune system activation ex vivo, as exemplified by the changes in the gene expression profile or immune cell infiltration during treatment with DNA-PK-I or chemotherapy. As immunotherapy lacks valid and predictive peripheral biomarkers and lymphocytes play a key role in tumor killing in immunotherapy, another aim of the present study was to understand whether germline variants in DDR genes may suggest a new target for poor responders to anti-PD-L1 drugs. To date, no ATLAS of PBMC germline variants predicting immunotherapy response exists in the literature. We selected genes of interest using the DDR panel and found that there are variants in responding patients that may be of greater clinical importance in predicting response to immunotherapy. Although this was a small cohort, in view of the scientific evidence reported, we hypothesize that this genetic predisposition could interplay with additional acquired mutations within the tumor that may contribute to modulate response to immunotherapy. We can conclude that PBMCs may be a good tool for study of biomarkers of response to immunotherapy based on *cGAS/STING* mRNA levels in LC patients. On the other hand, the analysis of CXCL10 and CCL5 expression in serum together with the analysis of the germline DDR panel need future studies. Serum analysis should be extended to a larger population, especially to understand this large variability at baseline in patients that may be affected by multiple inflammation triggers.

## 5. Conclusions

In conclusion, we used for the first time PBMCs from LC patients during immunotherapy as a tool to investigate features of immune responsiveness. We found that cGAS/STING activation in PBMCs, mainly in cytotoxic lymphoid cells, together with rare germline variants in the DDR gene panel, may be a useful tool to predict response to immunotherapy in LC patients but requires further confirmation in a larger number of patients. Moreover, we found that NR LC patients with absence of rare germline variants in DDR genes are potential candidates for a rationale of using DDRi in clinical practice.

## Figures and Tables

**Figure 1 biomedicines-12-00809-f001:**
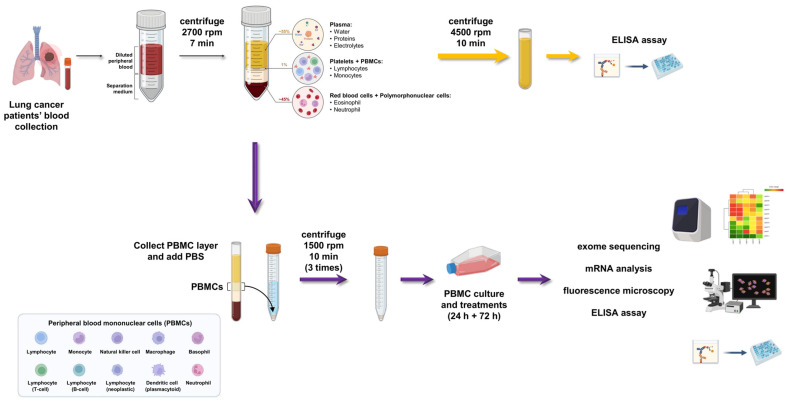
A graphical summary of the PBMC isolation method step-by-step. The graphical scheme was produced by the authors using the free BioRender platform.

**Figure 2 biomedicines-12-00809-f002:**
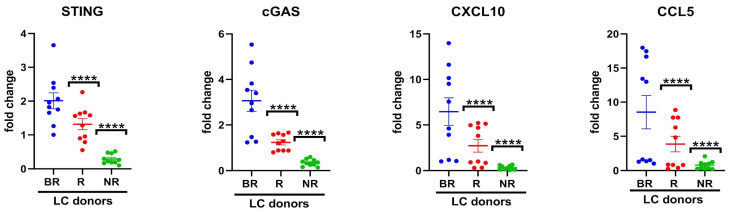
Real-time PCR analysis of in vitro *STING*, *c-GAS*, *CXCL10* and *CCL5* mRNA expression in lung cancer (LC) patient-derived PBMCs. Changes in mRNA levels were normalized to the expression of housekeeping genes (*18S*). Data are expressed as means ± SEM derived from *n* = 2 technical calculated by the comparative method 2^−∆∆Ct^. One-tailed unpaired Student’s *t*-test with CI = 95%, **** *p* < 0.0001.

**Figure 3 biomedicines-12-00809-f003:**
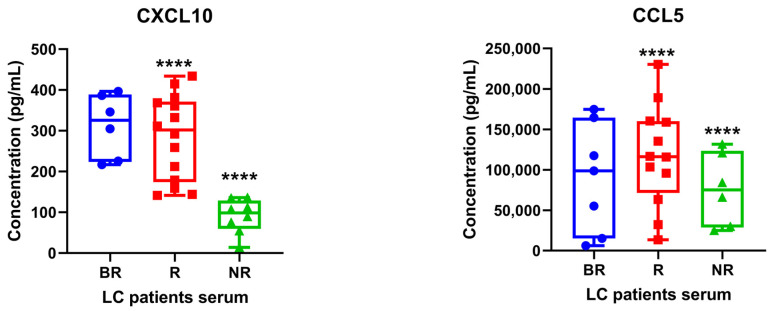
Levels of CXCL10/IP-10 and CCL5 in LC patients’ serum were determined using ELISA assay. Each experiment was performed in duplicate. Data represent mean concentrations ± SD. One-tailed unpaired Student’s *t*-test with CI = 95%, **** *p* < 0.0001.

**Figure 4 biomedicines-12-00809-f004:**
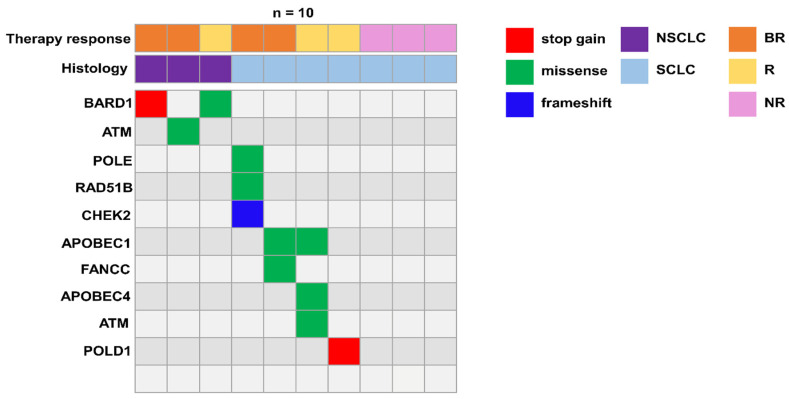
Gene name, variant types and number of variants of pathogenic DDR germline variants found in LC patients. Type and the therapy response of the corresponding LC groups are shown.

**Figure 5 biomedicines-12-00809-f005:**
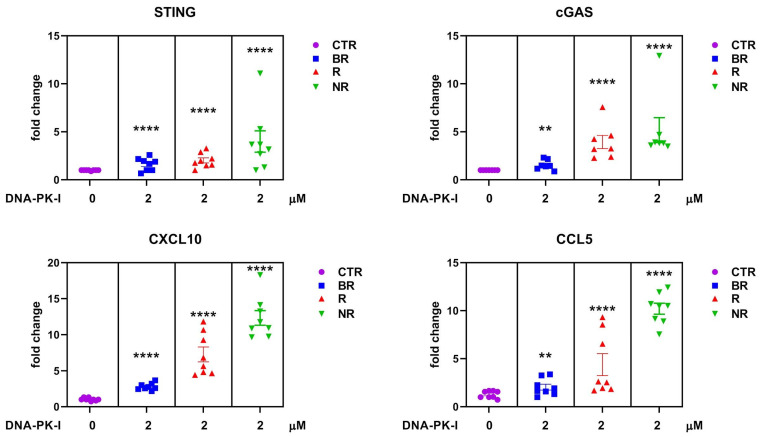
The mRNA expression levels of *STING*, *cGAS*, *CXCL10* and *CCL5* were studied by RT-qPCR in LC patient-derived PBMCs cultured in complete media supplemented with 10% HS for 24 h and then treated with DNA-PK-I 2 µM for 72 h; *18S* was used as the reference gene. Results are expressed as means ± SEM derived from *n* = 2 technical calculated by the comparative method 2^−∆∆Ct^. One-tailed unpaired Student’s *t*-test with CI = 95%, **** *p* < 0.0001, ** *p* < 0.05.

**Figure 6 biomedicines-12-00809-f006:**
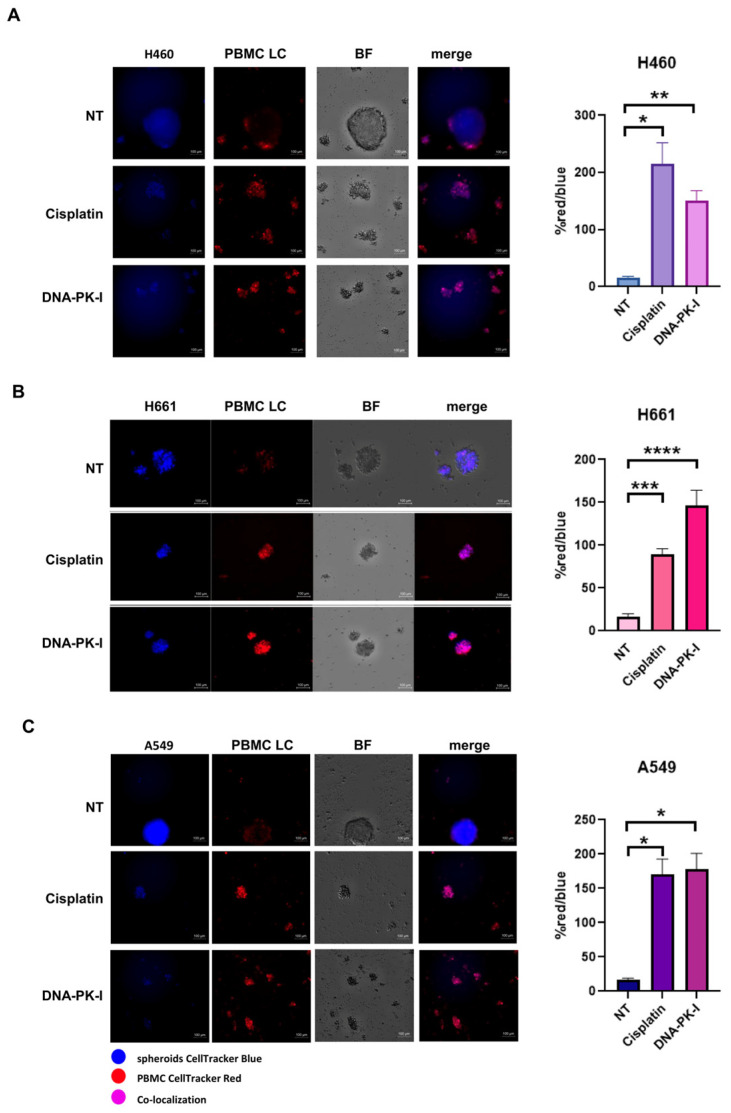
(**A**–**C**) Representative immunofluorescence images showing the infiltration ability of immune cells in 3D tumor spheroids of H460 (**A**), H661 (**B**) and A549 (**C**) cells and evaluation of the red/blue intensity ratio as a quantitative indication of the co-localization rate of LC cells and PBMC. Statistical significance: **** *p* < 0.0001, *** *p* < 0.001, ** *p* < 0.01, * *p* < 0.05. Three-dimensional tumor spheroids were blue stained, and the LC patient-derived PBMCs were red stained. The PBMCs were pre-treated with DNA-PK-I (2 µM) or cisplatin (0.5 µM) for 72 h and then added to 3D spheroids to start the co-culture. The co-localization of LC cells and PBMCs is reported under the merge column. Scale bar 100 µm.

**Figure 7 biomedicines-12-00809-f007:**
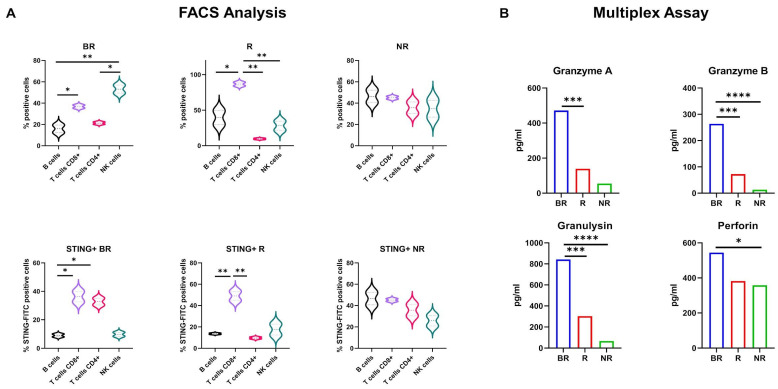
(**A**) FACS analysis of LC patient-derived PBMC subset (upper panel) and % of STING-positive cells (lower panel). Statistical significance: ** *p* < 0.001, * *p* < 0.05. (**B**) Luminex cytokine assay of granzyme A, granzyme B, granulysin and perforin in LC patient-derived PBMC culture supernatant. **** *p* < 0.0001, *** *p* < 0.001, * *p* < 0.05.

**Table 1 biomedicines-12-00809-t001:** List of primer sequences used for qRT-PCR.

Gene	Forward Sequence	Reverse Sequence
*18S*	5′-CGCCGCTAGAGGTGAAATTC-3′	3′-CTTTCGCTCTGGTCCGTCTT-5′
*STING*	5′-CAGGCACTGAACATCCTCCT-3′	3′-ATATACAGCCGCTGGCTCAC-5′
*cGAS*	5′-CTCCACGAAGCCAAGACCTC-3′	3′-GCGGCTGAGCTTCAACTTCT-5′
*CCL5*	5′-CTCCCCATATTCCTCGGACA-3′	3′-CTCTGGGTTGGCACACACTT-5′
*CXCL10*	5′-GGTGAGAAGAGATGTCTGAATCC-3	3′-GTCCATCCTTGGAAGCACTGCA-5′

**Table 2 biomedicines-12-00809-t002:** Patient characteristics.

	1L-IO (*n* = 3)	ICT (*n* = 25)
Samples (*n*)		
BR R NR	2 1 -	5 12 8
Age (mean, range)	66.00 (60–75)	66.68 (52–83)
Sex (*n*, %)		
Female Male	- 3 (100.00%)	6 (24.00%) 19 (76.00%)
Histology (*n*, %)		
SCLC NSCLC	- 3 (100.00%)	15 (60.00%) 10 (40.00%)

BR: best responders; R: responders; NR: non-responders; 1L-IO: patients receiving PD-L1 inhibitors as monotherapy in the first line; ICT: patients receiving chemotherapy plus PD-L1 inhibitors in the second or subsequent lines; 1L: first line; IO: immunotherapy. SCLC: small-cell lung cancer; NSCLC: non-small-cell lung cancer.

**Table 3 biomedicines-12-00809-t003:** Correlation analysis in lung cancer patients (BR, R and NR) between *STING/cGAS* and *CXCL10* and *CCL5* mRNA levels.

	BR					
	STING vs. CXCL10	STING vs. CCL5	STING vs. cGAS	cGAS vs. CXCL10	cGAS vs. CCL5	CXCL10 vs. CCL5
r value	0.6705	0.5582	0.3118	0.4053	0.1737	0.8777
95% CI	0.07070 to 0.9142	−0.1102 to 0.8789	−0.3955 to 0.7869	−0.3012 to 0.8245	−0.5119 to 0.7241	0.5545 to 0.9708
*p* value	0.0339	0.0936	0.3805	0.2452	0.6314	0.0008
	R					
	STING vs. CXCL10	STING vs. CCL5	STING vs. cGAS	cGAS vs. CXCL10	cGAS vs. CCL5	CXCL10 vs. CCL5
r value	0.4070	0.4382	0.7053	0.8643	0.8796	0.9051
95% CI	−0.2994 to 0.8252	−0.2643 to 0.8369	0.1362 to 0.9244	0.5149 to 0.9675	0.5602 to 0.9713	0.6405 to 0.9776
*p* value	0.2431	0.2052	0.0227	0.0013	0.0008	0.0003
	NR					
	STING vs. CXCL10	STING vs. CCL5	STING vs. cGAS	cGAS vs. CXCL10	cGAS vs. CCL5	CXCL10 vs. CCL5
r value	−0.1854	0.1827	0.3602	0.08317	0.5195	−0.4806
95% CI	−0.7298 to 0.5030	−0.5050 to 0.7285	−0.3485 to 0.8068	−0.5767 to 0.6773	−0.1637 to 0.8659	−0.8523 to 0.2137
*p* value	0.6082	0.6134	0.3066	0.8193	0.1238	0.1598

**Table 4 biomedicines-12-00809-t004:** Mutated DDR gene names and number of pathogenic variants in corresponding patient’s ID of SCLC and NSCLC are shown. We found germline heterozygous variants in all genes reported in the table. Variant read frequency is expressed as absolute value.

Patient ID	Genes	HGVSP	Consequence	Sift Prediction	PolyPhen Prediction	Variant Read Frequency	ClinVar
NSCLC_1	BARD1	p.(Arg406Ter)	stop_gained	-	-	0.48421052	VCV000229677.13:pathogenic
NSCLC_2	ATM	p.(Ser333Phe)	missense_variant	deleterious	probably damaging	0.47340426	VCV000127471.16:uncertain significance
NSCLC_3	BARD1	p.(Asp673Tyr)	missense_variant	deleterious	probably damaging	0.55279505	VCV000482778.3:uncertain significance
SCLC_1	CHEK2 POLE RAD51B	p.(Asn105IlefsTer11) p.(Lys425Arg) p.(Lys243Arg)	frameshift_variant missense_variant missense_variant	- deleterious deleterious	- probably damaging probably damaging	0.19469027 0.46486485 0.16216215	- VCV000224587.7:uncertain significance -
SCLC_2	APOBEC1 FANCC	p.(Glu41Gln) p.(Asp195Val)	missense_variant missense_variant	deleterious deleterious	- probably damaging	0.53571427 0.556701	- VCV000134305.15:uncertain significance
SCLC_3	APOBEC4 ATM APOBEC1	p.(Arg183Trp) p.(Ser49Cys) p.(Pro108Ser)	missense_variant missense_variant missense_variant	tolerated deleterious deleterious	- possibly damaging -	0.43418014 0.5242967 0.50517243	- VCV000003048.16:uncertain significance -
SCLC_4	POLD1	p.(Gln53Ter)	stop_gained	-	-	0.44516128	-
SCLC_5	-	-	-	-	-	-	-
SCLC_6	-	-	-	-	-	-	-
SCLC_7	-	-	-	-	-	-	-

## Data Availability

The majority of data related to presented results are included in the materials and methods section of the paper. NGS raw data were generated at facility center Ames (Casalnuovo, Napoli) and are available from the corresponding authors C.M.D.C. or V.D.R. on request.

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
