# Peer review of "PBMCs as Tool for Identification of Novel Immunotherapy Biomarkers in Lung Cancer"

_biomedicines, 2024, doi:10.3390/biomedicines12040809_

Round 1

Reviewer 1 Report

Comments and Suggestions for Authors

The manuscript identifies PBMCs as biomarkers for response to immunotherapy in lung cancer, they specially focus on cGAS-STING signaling. I have some concerns:

1.     Using PBMCs as biomarkers to evaluate the effect of immunotherapy is reasonable in some way, since PBMCs are quiet heterogeneous, the results would be more meaningful if they could focus on a specific subset.  

2.     Some parts of the results should be moved to the introduction.

3.     Limited sample size is a major shortcoming of the article, the author should include more samples in their research.

4.     As mentioned above, the cell source of CXCL10 should be detected, and the mechanism study of cGAS-STING signaling in cytokine production is needed.

5.     Correlation analysis of serum cytokine level and cGAS-STING signaling activation should be done.

6.     Animal study should be used to evaluate the effect of DNA-PK Inhibitor in increasing effect of anti-PDL1

7.     In Figure 5, in NR patients, the CXCL10 expression is higher than the R patients, while Figure 3 showed opposite results, the discrepancy should be explained.

8.     The discussion section should further emphasize the value and novelty of the current study should be d here.

9.     Fonts size in figures should be identical.

Reviewer 2 Report

Comments and Suggestions for Authors

De Rosa et al set out to determine if cGAS-STING signaling in isolated PMBCs isolated from NSCLC and SCLC patients can be used as biomarkers for predicting immunotherapy response. This is a important questions and the authors proved some interesting data supporting this hypothesis. However, a few issues limit enthusiasm for this report.

Major Issues

Sample sizes are small. While oftentimes the authors cannot control the availability of samples, they should discuss this as a limitation in the discussion.

The NCI-H661 cell line was derived from a large cell carcinoma patient (Phelps, et al. JBC, 1996), making its inclusion in this study confusing. A larger panel of SCLC and NSCLC cell lines should be tested.

Minor Points

Reference missing on line 284

The discussion should include the applicability of these results to transformed SCLC which is derived from NSCLC.

Round 2

Reviewer 1 Report

Comments and Suggestions for Authors

All concerns resolved.

Reviewer 2 Report

Comments and Suggestions for Authors

All comments addressed. Thank you to the authors for their revision.